# Foliar Fertilization by the Sol-Gel Particles Containing Cu and Zn

**DOI:** 10.3390/nano13010165

**Published:** 2022-12-30

**Authors:** Beata Borak, Krzysztof Gediga, Urszula Piszcz, Elżbieta Sacała

**Affiliations:** 1Department of Mechanics, Materials and Biomedical Engineering, Faculty of Mechanical Engineering, Wroclaw University of Science and Technology, Smoluchowskiego Str. 25, 50-370 Wroclaw, Poland; 2Department of Plant Nutrition, Institute of Soil Science, Plant Nutrition and Environmental Protection, The Faculty of Life Sciences and Technology, Grunwaldzka Str. 53, 50-357 Wroclaw, Poland

**Keywords:** foliar fertilization, sol-gel particles, Cu, Zn, maize *Zea mays*, wheat *Triticum sativum*, rape *Brassica napus* L. var *napus*

## Abstract

Silica particles with the size of 150–200 nm containing Ca, P, Cu or Zn ions were synthesized with the sol-gel method and tested as a foliar fertilizer on three plant species: maize *Zea mays*, wheat *Triticum sativum* and rape *Brassica napus* L. var *napus* growing on two types of soils: neutral and acidic. The aqueous suspensions of the studied particles were sprayed on the chosen leaves and also on the whole tested plants. At a specific stage of plant development determined according to the BBCH (Biologische Bundesanstalt, Bundessortenamt und CHemische Industrie) scale, the leaves and the whole plants were harvested and dried, and the content of Cu and Zn was determined with the AAS (atomic absorption spectroscopy) method. The engineered particles were compared with a water solution of CuSO_4_ and ZnSO_4_ (0.1%) used as a conventional fertilizer. In many cases, the copper-containing particles improved the metal supply to plants more effectively than the CuSO_4_. The zinc-containing particles had less effect on both the growth of plants and the metal concentration in the plants. All the tested particles were not toxic to the examined plants, although some of them caused a slight reduction in plants growth.

## 1. Introduction

Plants are the main source of food for humans. To grow and develop, plants must absorb macro- and micronutrients from fertilizers, but most conventional chemical fertilizers are characterized by a poor utilization and uptake efficiency due to the loss of fertilizer components caused by leaching, volatilization and precipitation or adsorption onto the soil complex that leads to environmental pollution [1]. The solution to this problem may be the use of nanotechnology, and more specifically nanoparticles (NPs).

Nanotechnology is a relatively new, innovative and promising field of research that opens up many opportunities in medicine, pharmaceuticals, electronics and also agriculture in the field of crop improvement and food processing [2,3]. NPs have unique optical, physical and biological properties in comparison to their molecular and bulk counterparts. These properties result from their small size, large surface area and high reactivity [2,4,5], which enables better interactions and a more efficient uptake of nutrients by plants.

Nanotechnology and NPs could play a key role in agricultural production and securing food for a growing human population. In the context of adverse climate change and shrinking arable land and water resources, it is very important to use the right agricultural techniques that will allow one to further increase crop yields without a negative impact on the environment. The primary factor ensuring high yields is the proper fertilization of crops; hence, the use of NPs-based fertilizers (nanofertilizers) can lead to a more effective supply of nutrients to plants and their better use [3]. Mineral nutrients are not only required for better plant growth and development, but also are crucial in increasing plant resistance to different kinds of environmental stresses. Fertilizers containing micronutrients are also used to increase the concentration of needed elements in edible parts of crops in regions where there are dietary deficiencies in humans and livestock.

Nanofertilizers in the form of NPs can be applied to the soil or directly to plants as foliar sprays [6]. Soil application is more common and most effective for nutrients which are required in high amounts. However, foliar sprays are widely used in agricultural production as an alternative or complementary strategy to soil fertilization. Under certain circumstances, for example, a deficiency of a defined essential element, especially a microelement, foliar fertilization may be more economic and effective than a soil fertilizer. Foliar applications of some essential nutrients in the form of NPs can be used to supplement their deficiency in plants grown in the soil. However, there are some requirements for successful foliar fertilization. The process of the uptake, translocation and bioaccumulation of the nanofertilizer depends on the plant species, fertilizer characteristics and environmental conditions [7,8,9,10]. In order to absorb a sufficient amount of the applied nutrient solution, the plants should have a high leaf area index for absorbing the applied nutrient solution in sufficient amounts, and a fertilizer source should be water soluble or eventually form stable aqueous dispersion [11,12,13]. Nutrient concentration, molecular size and weight, electric charge, pH of the solution and point of deliquescence are factors that characterize the fertilizer and should be taken into account. Additionally, the environmental conditions such as day temperature and humidity should be optimal to avoid leaf injury, burning and other harmful effects. It is worth noting that the solution containing mineral fertilizers can be supplemented with postemergence herbicides, insecticides or fungicides. Such a solution is advantageous because of higher potential yield increases and reduced application costs [6].

Great attention should be paid to the NPs toxicity and their interaction with plants. The release of NPs to the environment could be potentially hazardous to living organisms and human health and life. Once entered in the soil system, nanomaterials may affect the soil quality and plant growth. NPs, connected with plants, enter the human food chain. Hence, there is a need to study the toxicological effects of NPs and their mobility [14].

Copper (Cu) and zinc (Zn) are important micronutrients essential for plant growth and development and are crucial not only for crop productivity but particularly crop quality all over the world [15,16]. The soils of many regions of the world, especially in dry regions, are often poor in plant-available Zn, which results in the so-called hidden zinc hunger among inhabitants whose diet is based on plant-based foods. Hence, Zn fertilization is expected to have positive effects on human health, especially in areas of Zn malnutrition [17].

Currently, there is a significant interest in using nanotechnology to correct the deficiency of some essential nutrients, especially metallic microelements such as Fe, Cu, Zn and Mn [18]. Two of them (Cu and Zn) were the subject of this research and have been introduced into SiO_2_ particles and used as foliar fertilizer sprayed onto wheat, maize and rape leaves. Wheat (*Triticum aestivum* L.) is one of the most consumed cereals and occupies the largest area in the world. It is followed by maize and rice. Oilseed rape is widely cultivated in Poland. An optimal supply of nutrients to plants is crucial for proper plant growth and development, high productivity and tolerance to both biotic and abiotic stress factors and the health of plant-eating organisms.

The aim of our research was to (i) examine silica particles containing Cu or Zn as potential foliar fertilizers, (ii) compare the silica particles’ effect on plants with their conventional equivalent and (iii) assess their possible toxicity to three investigated plants: maize, wheat and rape.

## 2. Materials and Methods

### 2.1. Chemicals

Tetraethoxysilane TEOS (99.9% Alfa Aesar Gmbh&Co KG, Karlsruhe, Germany), ethanol EtOH (96%, POCH, Gliwice, Poland), ammonium hydroxide solution (25%, POCH, Gliwice, Poland), nitric acid HNO_3_ (65%, POCH, Gliwice, Poland), ortho-phosphoric acid H_3_PO_4_ (85%, POCH, Gliwice, Poland), calcium nitrate tetrahydrate Ca(NO_3_)_2_·4H_2_O (Chempur, Piekary Śląskie, Poland), copper(II) nitrate hemi-(pentahydrate) Cu(NO_3_)_2_·2.5H_2_O (Alfa Aesar Gmbh&Co KG, Karlsruhe, Germany) and zinc nitrate hexahydrate Zn(NO_3_)_2_·6H_2_O (Chempur, Piekary Śląskie, Poland) were used for the particle synthesis. Copper sulfate CuSO_4_ and zinc sulfate ZnSO_4_ in the form of a 0.1% solution were used as a source of Cu^2+^ and Zn^2+^ ions in the fertilizer applications.

### 2.2. Particle Synthesis

Particles based on SiO_2_, CaO, with/without P and doped with Cu and Zn (marked as BG) were synthesized using the sol-gel technique through a modified Stöber method carried out in a two-step acid–alkaline process [19]. The acidic step of the synthesis involved the preparation of a TEOS dispersion (1.5 mL) in 6 mL of EtOH and 5 mL of H_2_O in the acidic environment generated by adding 1 µL of HNO_3_ (samples without P) or 73 µL of H_3_PO_4_ as a P precursor in the samples with P. The basic reaction step involved preparing a solution consisting of 25 mL of EtOH, 5 mL of H_2_O and 7.5 mL of NH_3_. As sources of Ca, Cu and Zn, nitrates were used, which, after dissolving in 4 mL of H_2_O, were added to the previously mixed acidic and alkaline solutions. The tested samples differed in their chemical composition (presence or absence of P) and the content of Ca and Cu or Zn. Details are presented in Table 1, where the mass of these metals is given in relation to the 1 kg of the material.

The BGCu sample was obtained in a slightly different way. A total of 3 mL of TEOS was dissolved in 25 mL of EtOH and added to the solution, which consisted of 10 mL of EtOH, 5 mL of H_2_O and 4.5 mL of NH_3_. To the same solution, calcium and copper nitrates dissolved separately in 5 mL of H_2_O were also added.

After 24 h of stirring, the obtained precipitates were separated from the reaction mixture by centrifugation and were washed three times with deionized water. All the synthesized samples were annealed for 2 h at the temperature of 700 °C in the air, and the final white powders were ground in a mortar before being used for plant tests.

The SiO_2_/Cu^2+^ sample was obtained by impregnation of the previously obtained SiO_2_ particles in the solution of Cu(NO_3_)_2_·2.5H_2_O (0.24 M). After 1 h of mixing SiO_2_ with the copper solution in an ultrasonic scrubber, the sample was separated by centrifugation and dried at room temperature.

The morphology and composition of the obtained samples were analyzed with scanning electron microscopy (SEM S-3400N HITACHI) equipped with an energy-dispersive X-ray (EDS) detector. The parameters selected in order to obtain good quality images of the samples were: accelerated voltage 10 kV, emission current 67–74 µA and working distance in the range 9–11 mm. For the measurements, the samples were put on carbon tape and covered with a carbon layer to avoid charging effects. The size of the particles was analyzed using a DLS (Dynamic Light Scattering) technique and Nano-ZS Malvern Instruments after ultrasonification of the powders in distilled water. Three measurements were made for each material. The amorphous or partially crystallized structure of the samples was analyzed by X-ray diffraction measurements (ULTIMA IV/Rigaku/2008), using Cu Kα (λKα = 1.5406 Å) radiation in the 2θ range from 5 to 70°.

### 2.3. Plant and Soil Materials

In the study, three plant species were tested: maize *Zea mays*, wheat *Triticum sativum* and winter rape *Brassica napus* L. var *napus* growing on two types of soils: neutral pH in 1 mol KCl = 7.1 with the medium content of available copper and zinc and acid pH = 5.6, with a low content of available copper and zinc.

### 2.4. Foliar Application

Aqueous suspensions (0.1% *m*/*v*) of all the tested materials were prepared with a high-purity water Type I (0.055 µS cm^−1^). The aqueous suspensions of the tested particles were sprayed on the chosen leaves and also on the whole tested plants. At a specific stage of development determined according to the BBCH scale (“Biologische Bundesanstalt, Bundessortenamt und Chemische Industrie”), the leaves and the whole plants were harvested and dried, and the concentrations of Cu and Zn were determined using the atomic absorption spectroscopy (AAS) method with a SpectrAA 220FS spectrometer (Varian, Melbourne, Australia) following dry mineralization in a muffle furnace at 450 °C and dissolving the ash in 1 mol dm^−3^ nitric acid (Merck, KGaA, Darmstadt, Germany). The control sample was the plant sprayed with water.

The BBCH scale is a system for uniform coding phenologically similar growth stages of all mono- and dicotyledonous plant species [20,21]. It is a widely used scale for standardized descriptions of plant development stages in order of their phenological characteristics. The BBCH scale provides uniform criteria for the description, identification and selection of the phenological stages of the plants.

The samples of wheat were treated with the particles’ suspension in BBCH21 at the step of the beginning of tillering when the first tiller was detectable. The wheat plants were harvested in BBCH30 at the beginning of stem elongation. The maize samples were fertilized in the phase of 4 leaves unfolded (BBCH14) and harvested in the phase of 8 leaves unfolded (BBCH18). The rape samples were sprayed with the particles’ suspension in BBCH14 (4 leaves unfolded) and harvested in BBCH 18 (9 leaves unfolded).

### 2.5. Statistical Analysis

Differences among means in the experimental data were analyzed with a two-way ANOVA for the homogeneous groups, and the post-hoc Tukey HSD test was also performed. All these analyses were carried out at the level *p* = 0.05 with Statistica software version 13. (TIBCO Software Inc. (2017). Statistica (data analysis software system), version 13. http://statistica.io (accessed on 20 December 2022) under a permanent license for Institution). The figures were made with QtiPlot 1.0.0-rc15 (64-bit) software under a permanent license released on 31 August 2021.

## 3. Results

### 3.1. Sol-Gel Particles Characterization

All the tested particles were synthesized by the sol-gel process based on the modified Stöber method [22]. It is a common way to obtain silica SiO_2_ spherical particles but some necessary adaptations were introduced to obtain particles based on SiO_2_ and also containing Ca, P_,_ Cu or Zn. The synthesis process consisted of two steps: acidic, in which the TEOS–silica precursor was hydrolyzed to sol, and alkaline, in which the condensation to gel and precipitation of the particles took place [23]. The heat treatment of the samples (700 °C) was needed to incorporate calcium and other dopants into the silica structure and to remove organic residues and nitrate byproducts [24].

#### 3.1.1. DLS Analysis

The size of the synthesized particles was analyzed with the DLS method. The obtained data and PdI (polydispersity index) parameters are presented in Table 2. Graphs showing the determined particle sizes are in the Appendix A.

The size of the synthesized particles was in the range of 150–200 nm. Two samples with particles size of about 200 nm (BGCu and SiO_2_Cu) were synthesized following the typical Stöber synthesis in alkaline conditions. Other samples, with smaller particle sizes (150–180 nm), were synthesized in a two-step acid–alkaline process. No clear effect of P or the introduction of Cu and Zn on the particle size was observed. Samples with a higher amount of Zn (BGZn15 and BGPZn15) had slightly larger particles than the samples with a smaller amount of Zn (BGZn10 and BGPZn10).

#### 3.1.2. SEM and Energy-Dispersive X-ray (EDX) Analysis

A surface analysis of the synthesized particles was performed via SEM observation. The chemical composition of the samples was assessed using an EDX analysis. The presence of silicon (Si), calcium (Ca), phosphorus (P)—if added—and the dopants copper (Cu) or zinc (Zn) was clear. Figure 1 and Figure 2 present the SEM micrographs and Table 3 and Table 4 present the EDX analysis of the particles doped with Cu and Zn, respectively. The studied samples differed in their chemical composition (presence or absence of P) and the content of Cu and Zn. The particles doped with Cu had a size in the range of 200–250 nm (Figure 1), and the particles doped with Zn seemed to be slightly smaller, with a size of about 200 nm or less (Figure 2). The particles were almost spherical in shape. After heat treatment at the temperature of 700 °C, the particles did not undergo co-sintering. The particles of the BGPZn10 and BGPZn15 samples seemed to be smaller than the others, but the DLS analysis did not confirm this. 

The particle sizes determined by DLS and SEM were similar. The spheres visible in the SEM images were not separated; hence, we were not accurate when determining their size.

#### 3.1.3. X-ray Diffraction (XRD)

The XRD patterns of the particles doped with Cu are shown in Figure 3. The absence of sharp diffraction peaks in the samples without P, i.e., BGCu and BGCu10, indicated that these samples were amorphous. The broad peak with a maximum around 22.5° was attributed to the SiO_2_ amorphous halo, whereas the small protrusion at 30° observed at the XRD pattern for BGCu and BGCu10 was due to the incipient crystallinity of dicalcium silicate beta Ca_2_SiO_4_ (ICSD 16616) (Figure 3A) [23]. The sample with Cu and P (BGPCu15) (Figure 3B) was crystalline and consisted of the tricalcium tricopper tetrakis(phosphate(V)) Ca_3_Cu_3_(PO_4_)_4_ (ICSD15519) and other crystalline phases that were not identified.

The XRD patterns of the samples doped with Zn are shown in Figure 4. Samples without P (BGZn10 and BGZn15) showed the presence of the crystalline phase, which was identified as a dicalcium silicate beta CaSiO_3_ (ICSD 16616). The samples containing P (BGPZn15 and BGPZn10) were clearly crystalline (Figure 4B), and the presence of calcium zinc phosphate oxide hydroxide Ca_10_Zn_0.30_(PO_4_)_6_(O_0.68_(OH)_1.24_) (ICSD 183743) was identified. Other crystal phases were also presented but could not be identified. The samples containing Cu and Zn but without P showed traces of the crystalline calcium silicate, while the same samples with P were evidently crystalline. It is clear that the presence of P in the material structure initiated/accelerated the crystallization process. It is a typical phenomenon observed for this kind of material [24].

### 3.2. Foliar Fertilizer Application

The synthesized particles containing Zn or Cu (abbreviations BG, Table 1), silica particles impregnated in a solution of copper nitrate (SiO_2_/Cu^2+^) and solutions of Cu^2+^ or Zn^2+^ ions (*w*/*v* 0.1%) in the form of sulfates (copper sulfate and zinc sulfate) were investigated as potential foliar fertilizers.

For foliar applications, the synthesized particles were homogenously dispersed into water via ultrasonification and formed a stable suspension with negligibly low precipitation. The aqueous suspensions were used to spray the tested plants at their specific development stage according to the BBCH scale (point 2.5). In the case of wheat, the whole aboveground parts were sprayed. In the case of rape and maize, two different modes of application were used. One selected leaf or the whole rape and maize plant were sprayed. The control plants were sprayed with water. The tested plants were cultivated in the soil with a different pH (acidic or neutral), affecting changes in bioavailability and the uptake of metals present in the soil. At the end of the growing season, the plants were harvested (according to their developmental stage and the BBCH scale) and the concentration of Cu and Zn was determined either in that particular leaf (maize or rape) or in the whole plant (wheat, maize or rape). This approach was aimed at obtaining some information regarding the movement of the examined microelements in the plants. The plants’ yields (dry mass per pot) were also determined.

#### 3.2.1. Foliar Fertilizer Applied on Wheat (*Triticum sativum*)

Wheat treated with BGCu was characterized by the significantly lowest dry mass yield, particularly in the acidic soil (Figure 5). Under other treatments, some differences in wheat yield were observed; however, they were not statistically significant. The preparations containing Zn had no significant effect on wheat growth (Figure 5). Under tested conditions (different formulations used and the type of soil), the Cu concentrations in the aboveground part of the wheat were in the range of 5.5–9.5 mg kg^−1^ D.M., with the highest concentration of 9.5 mg kg^−1^ D.M. recorded in plants growing in acidic soil and sprayed with BGCu. In contrast, wheat growing on neutral soil had the lowest Cu content. Generally, the plants that grew in the neutral soil accumulated less Cu than those growing in the acidic soil (Figure 5). An exception was the control plants, in which the Cu content was the same. A similar tendency was observed for the Zn concentration. The acidic soil promoted Zn accumulation in the wheat, except for the control plants.

#### 3.2.2. Foliar Fertilizer Applied on Maize (*Zea mays*)

Different treatments caused some changes in dry matter production. The plants from the objects treated with BGCu10, BGCu15, SiO_2_/Cu^2+^ and copper sulfate had significantly low yields compared to the control and BGCu grown on neutral soil (Figure 6). In the case of Zn-containing compounds, a significant decrease was observed in the plants grown on acidic soil under BGZn10, BGZn15 and BGPZn15 treatments. Generally, the dry mass of maize was significantly higher from pots with acidic soil.

Maize grown on acidic soil had a low copper content per whole plant (Figure 7A,C). On the other hand, the plants grown on neutral soil (except for the plants sprayed with water—the control plants) had a slightly higher Cu content per the whole plant than the plants grown in acidic soil (Figure 7A,C). Extremely high Cu content reaching 52 mg kg^−1^ D.M. was found in the plants treated with copper sulfate. Two other products (BGPCu15 and BGCu) also caused an increase in the copper content compared to the control plants, but the effect was considerably lower than for sulfate. The recorded concentration was 10 mg kg^−1^ D.M. (Figure 7B). The Cu content determined in the individual treated leaves was notably higher than in the whole aboveground part of the plants, and the effect of the soil pH was mostly not significant (Figure 7B,D). The Zn content was lower in the individual examined leaves than in the whole aboveground part of the plant (Figure 8B,D). The exceptions were the plants growing on neutral soil and treated with zinc sulfate. They accumulated huge amounts of zinc in their tissues, with concentrations reaching up to 180 mg kg^−1^ D.M. in the examined leaves. The dry matter production was greater in the acidic soil than in the neutral soil, and three preparations resulted in a lower dry mass (Figure 6). For the neutral soil, the examined preparations did not have a statistically significant effect on this value.

#### 3.2.3. Foliar Fertilizer Applied on Rape *Brassica napus* L. var *napus*

The yield of rape treated with Zn-containing formulations and grown both on acidic and neutral soils was relatively stable (Figure 9). In the case of Cu-containing formulations, relatively little changes in the dry matter production were observed in the plants grown on neutral soil. On acidic soil, the BGCu and BGCu 10 formulations caused a marked decrease in the dry matter of rape (Figure 9). In general, the weakest-acting formulation was BGCu, which also reduced dry matter production on acidic soil (Figure 9). A similar effect was observed in wheat.

In the case of rape, the Cu concentration in the control plants was about 5.5 mg kg^−1^ D.M., and this value was lower than the optimal level for most crops (Figure 10A,C). The plants treated with the examined formulations (excluding BGCu) were better supplied with copper than the control plants, and this effect was especially pronounced on acidic soil. The highest concentration of Cu was found under the influence of SiO_2_/Cu^2+^ (Figure 10A–D). The relatively high concentration of Cu in the individual leaves tested may suggest that the introduced copper was difficult to move within the plant (Figure 10B,D).

The zinc content in the rape, both in individual leaves and the whole plant, remained high (Figure 10). In some cases, particularly in acidic soil, the Zn concentrations exceeded 50 mg kg^−1^ D.M., while the optimal range is 30–100 mg Zn kg^−1^ D.M. The highest Zn concentration was observed in the rape treated with zinc sulfate regardless of soil pH.

## 4. Discussion

Silica NPs loaded with a drug or other active substance have been studied as smart delivery systems for various applications in medicine and pharmacology [25]. The same idea could be transferred to agriculture and applied to plants to provide fertilizers (macro- and microelements) and protective substances (pesticides or herbicides) [3].

In the presented work, silica particles containing Ca, P (or without), Cu or Zn ions were tested as a foliar applied fertilizer. Particles without P showed traces of the crystalline form of dicalcium silicate and amorphous SiO_2_. Since the XRD spectra of the samples heated at 700 °C did not show reflections from oxides or other salts containing Cu and Zn, it can be assumed that these ions were randomly dispersed in the network formed by SiO_2_. The particles containing P were crystalline after calcination at 700 °C. Tricalcium tricopper tetrakis(phosphate(V)) Ca_3_Cu_3_(PO_4_)_4_ in the samples with Cu and calcium zinc phosphate oxide hydroxide Ca_10_Zn_0.30_(PO_4_)_6_(O_0.68_(OH)_1.24_) in the samples with Zn were detected on the XRD diffractograms. As a foliar fertilizer, silica particles were also impregnated in the solution of copper nitrate (SiO_2_/Cu^2+^), and 0.1% solutions of copper sulfate and zinc sulfate were tested. In these samples, metals were in the form of ions Cu^2+^ and Zn^2+^ dissolved in an aqueous solution.

Data concerning the effect of nanoparticles on plants indicate that the plant response depends on various factors such as plant species, the size and shape of the particles, their stability and the metal concentration [7,8,9,10]. Tested metals (Cu, Zn) were assimilated by the plants in a different way. The uptake of these metals was influenced by plant species, soil pH, a form of metal (water solution or water suspension of particles) and particle size. The metal content in the sample had an ambiguous effect. The wheat grown on neutral soil showed a lower Cu concentration than that grown on acidic soil with the exception of control plants (Figure 5). In the case of the plants grown on acidic soil, a little increase in the Cu content was observed under the influence of all the used spray variants. The preparations containing Zn had no significant effect on wheat growth, regardless of the pH of the soil (Figure 5). Cu and Zn applied as a foliar fertilizer in the form of ions in an aqueous solution (SiO_2_/Cu^2+^, copper sulfide and zinc sulfide) did not cause a significant increase in the content of these metals in the wheat in comparison to other samples, in which they formed phosphates (samples with P) or were distributed in the SiO_2_ network (BGCu, BGCu10, BGZn10 and BGZn15). It might seem that metals in the form of an aqueous solution should be easily accessible to plants, but in the case of wheat, this was not confirmed. In general, in the case of wheat, there was no clear effect of the tested material on the differentiation of the content of both metals in the plant, and in this aspect, wheat showed little susceptibility to foliar fertilization.

In many cases, the tested preparations containing copper (BGCu, BGCu10, BGPCu15, SiO_2_/Cu^2+^, solution of copper sulfate) effectively increased the level of this micronutrient in the maize and rape grown on both acidic and neutral soil (Figure 7 and Figure 10). A particularly high Cu content was observed in the maize (the whole plant) grown on neutral soil and treated with an aqueous solution of copper sulfate (Figure 7C). A high Cu content was also present in the leaves of a plant growing on both neutral and acidic soil and treated with SiO_2_/Cu^2+^ (Figure 7B,D and Figure 10B,D). This may indicate the relatively low mobility of the examined particles. The higher content of Cu in the tested preparations and the presence of P (which means the crystalline form of the sample) did not affect the greater assimilability of this element. Samples with different contents of Cu and containing P did not stand out among the others (Figure 7 and Figure 10). A different maize response was observed with respect to the preparations providing Zn. The Zn content was lower in the individual examined leaves than in the whole aboveground part of the plant. It might suggest that the Zn from these nanoparticles was transported to other plant tissues. The higher content of Zn in the plant was clearly visible after the application of the preparation in the form of an aqueous solution of zinc sulfate (Figure 8). The effect of the other preparations was insignificant. Additionally, in most cases, the content of Zn was similar to that of the control sample. The pH of the soil had no significant effect on the uptake of Zn. Formulations containing P were as effective as the other nanoparticles.

In most cases, the formulations containing Cu effectively increased the Cu level in the individual leaf and the whole plant rape grown on both acidic and neutral soil (Figure 10). Similarly to maize, the transport of Cu from the sprayed leaf to the remaining tissues of the plant seemed to be limited. The weakest effect on the increase in the Cu content in the tested plants was shown by the BGCu preparation having the largest particles (200 nm) and containing the least Cu. In the case of Zn given to rape, a high content of this element was observed in the leaves after treatments with zinc sulfate (Figure 11). Generally, the zinc concentrations in the individual leaves were equal to or lower than in the whole plants (except for zinc sulfate), and this might suggest that metal can move within the plant. There was no clear effect of a higher content of Zn in the preparation or the presence of a crystalline compound in the samples with P on the effectiveness in terms of both the effect on plant growth and zinc accumulation in plant tissues. Our experiments did not directly examine the uptake, assimilation and transport of metals in NPs; however, some effects related to metal mobility were noticeable. The behavior and fate of NPs within plants and their mechanism of action are still not fully understood.

Foliar-applied NPs can enter plant leaves through the pores of stomata, cuticular nanopores or the bases of trichomes [3,15,26]. For uptake and translocation, NPs must overcome a number of chemical and physiological barriers, with particle size being the important limiting factor. The tested particles were in the size range of 150–200 nm and they were larger than the size of pores in cell walls (5–20 nm) and plasmodesmata (3–50 nm). However, their size is just right to enter through stomata, hydathodes, lenticels, pit membranes and cell membranes. Stomates are microscopic pores in the epidermis of the leaf, and when they are opened, foliar absorption is easier. Plants species vary in the number of stomates per leaf area and their relative distribution. Some plants have more stomates on the lower leaf surface than on the upper and some vice versa [15]. Dicotyledon plants (rape) usually have more stomata on the lower surface of the leaves than the upper surface, whereas wheat and maize have about the same number of stomata on both leaf surfaces. It is worth mentioning that although the entry and uptake of NPs are limited by their size, larger particles were observed in tissues where theoretically they should not be present [3,10]. The particles are likely to induce the formation of new and larger pores in cell walls and cuticles or cause structural changes (e.g., microfilament ruptures and disruption), which in turn facilitates their absorption. The mechanism of the foliar absorption of NPs is not well understood. It has also been shown that environmental factors such as coexisting organic and inorganic compounds, soil salinity, drought, nutrient availability, environmental pollutants and others can affect particle uptake by plants [10].

Extensive literature data indicate both the beneficial and toxic effects of NPs. NPs have shown a positive effect on plant growth and yield [15,27,28,29], but available reports have also shown that they can be toxic to plants [30,31,32,33]. The use of metals in plant fertilizers, especially heavy metals, requires a precisely selected dose because metals, even those necessary for the normal growth and development of plants, after exceeding the tolerance threshold, will have a toxic effect on metabolism and lead to the inhibition of plant growth.

SiO_2_ was the main component of the tested materials, and it acted as a carrier in which Cu and Zn ions were closed. Some scientific reports have indicated the negative effect of SiO_2_ nanoparticles on plants; it was shown that the introduction of SiO_2_ NPs into the nutrient medium resulted in a significant decrease in cotton growth parameters [34]. On the other hand, there are also reports indicating the positive effect of SiO_2_ NPs on plants, especially those exposed to environmental stresses, such as salinity [35] or drought [36,37]. It was observed that SiO_2_ NPs increased biomass and enhanced the fruit yield of watermelon [38] and showed no acute toxic effects in wild pear irrigation [39]. The positive effect of nanosilica on maize crops was found in various aspects: it increased the seed germination capacity, water use efficiency and total chlorophyll content [40]. Some plant systems accumulate silica in solid form, creating intracellular or extracellular silica bodies (phytoliths) [41]. The phytotoxicity of NPs and SiO_2_ depends on their characteristics, such as particle size, concentration and surface composition [42,43]. Generally, Si is recognized as an element that is beneficial to plants, which promotes their growth and is not harmful even in high concentrations [44,45]. However, its application as SiO_2_ requires further research.

The studied particles were safe for the examined plants and no symptoms of toxicity (chlorosis, necrosis, leaf curling or developmental disorders) were recorded during the experiment. In a few cases, a reduction in the plant dry weight was found (Figure 5, Figure 6 and Figure 9) and this effect was more pronounced in the plants grown on acid soil. This unfavorable effect was observed only under certain conditions (plant, soil). BGCu caused a decrease in the dry matter of the wheat and rape grown on acidic soil. On the other hand, BGPCu15 and SiO_2_/Cu^2+^ reduced the growth of maize cultivated on neutral soil (Figure 6). The obtained results clearly indicate that the reduction in dry weight was not caused by excessive copper accumulation in the aboveground parts of the plants; it rather seems that it may be due to the physicochemical properties of the particle itself and its specific interaction with the plant. The content of Cu derived from BGCu in the tested plants was at the same level as the content of this element in the control samples (except for the wheat grown on acidic soil), but it was also lower than that observed for the other Cu-containing samples (BGCu10, BGPCu15, BG/Cu^2+^). What distinguishes the BGCu sample is its size. The DLS measurement showed that the BGCu particles had a diameter size of about 200 nm, and they were bigger than the other ones doped with Cu and Zn. BG/Cu^2+^ also had a similar size, but in this case, Cu was loaded into silica particles via impregnation with the CuSO_4_ solution, not by calcination.

It is worth noting that the other preparations containing copper (BGCu10, BGCu15 and SiO_2_/Cu^2+^) were effective in increasing the level of this micronutrient in rape grown on both acidic and neutral soil. Copper sulfate had a similar effect. In the case of the wheat grown on acidic soil, an increase in the metal content was observed under the influence of all the spray variants used, including copper sulfate. On neutral soil, BGCu caused a decrease in the copper concentration in wheat tissues, and the other sprays had no effect. All the compounds tested as foliar sprays did not significantly change the copper concentration in the aboveground parts of the maize grown on the acid soil. In contrast, in the neutral soil, the greatest increase in the Cu concentration was found in the presence of copper sulfate, although the treatment of the aboveground parts of the maize with BGCu15 and SiO_2_/Cu^2+^ also contributed to the increased levels of the metal in the plant (Figure 7).

It is obvious that nutrient assimilation under controlled conditions in the laboratory can be completely different from what is observed in real field conditions. Hence, the direct extrapolation of the results from the laboratory to the field requires more studies. Our goal was to investigate in general whether synthesized NPs have any positive effect, and further detailed research should test the toxicity of all particle components.

## 5. Conclusions

This study describes eight different engineered silica particles containing Cu or Zn that differ in composition and structure. Samples containing Si, Ca, P and Cu or Zn were crystalline, and the same samples without P showed traces of the crystalline form of calcium silicate. The studied particles had spherical shapes and diameters in the range of 150–200 nm. All the engineered particles were tested as potential foliar fertilizers and compared with conventional products—a water solution of ZnSO_4_ and CuSO_4_. The examined plants maize *Zea mays,* rape *Brassica napus* L. var *napus* and wheat *Triticum sativum* differed in their response to the same particles. In many cases, copper-containing particles improved the metal supply to plants more effectively than the copper sulfate used. Zinc-containing particles had less effect on both the growth of plants and the concentration of the metal in the plants. All the tested particles were not toxic to the plants examined, although some of them caused a slight reduction in plants growth.

The use of nanoparticles as a foliar fertilizer has great potential, but formulations must be targeted to specific crops and specific growing conditions to achieve the desired results. Based on our data and other studies, we believe that further research should focus on tracing the fate of metals encapsulated in nanoparticles, their release and movement in the plant and their localization in the cell.

## Figures and Tables

**Figure 1 nanomaterials-13-00165-f001:**
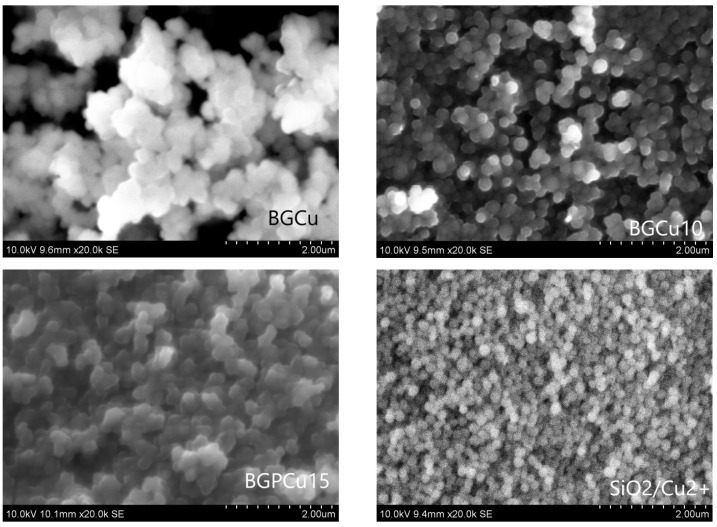
SEM micrographs of particles doped with Cu: BGCu, BGCu10, BGPCu15 and SiO_2_ particles impregnated in a solution of copper nitrate (SiO_2_/Cu^2+^).

**Figure 2 nanomaterials-13-00165-f002:**
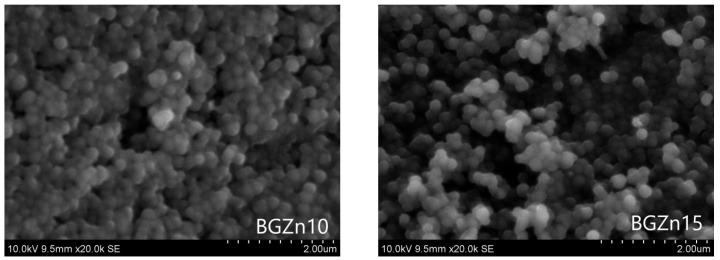
SEM micrographs of particles doped with Zn: BGZn10, BGZn15, BGPZn15 and BGPZn10.

**Figure 3 nanomaterials-13-00165-f003:**
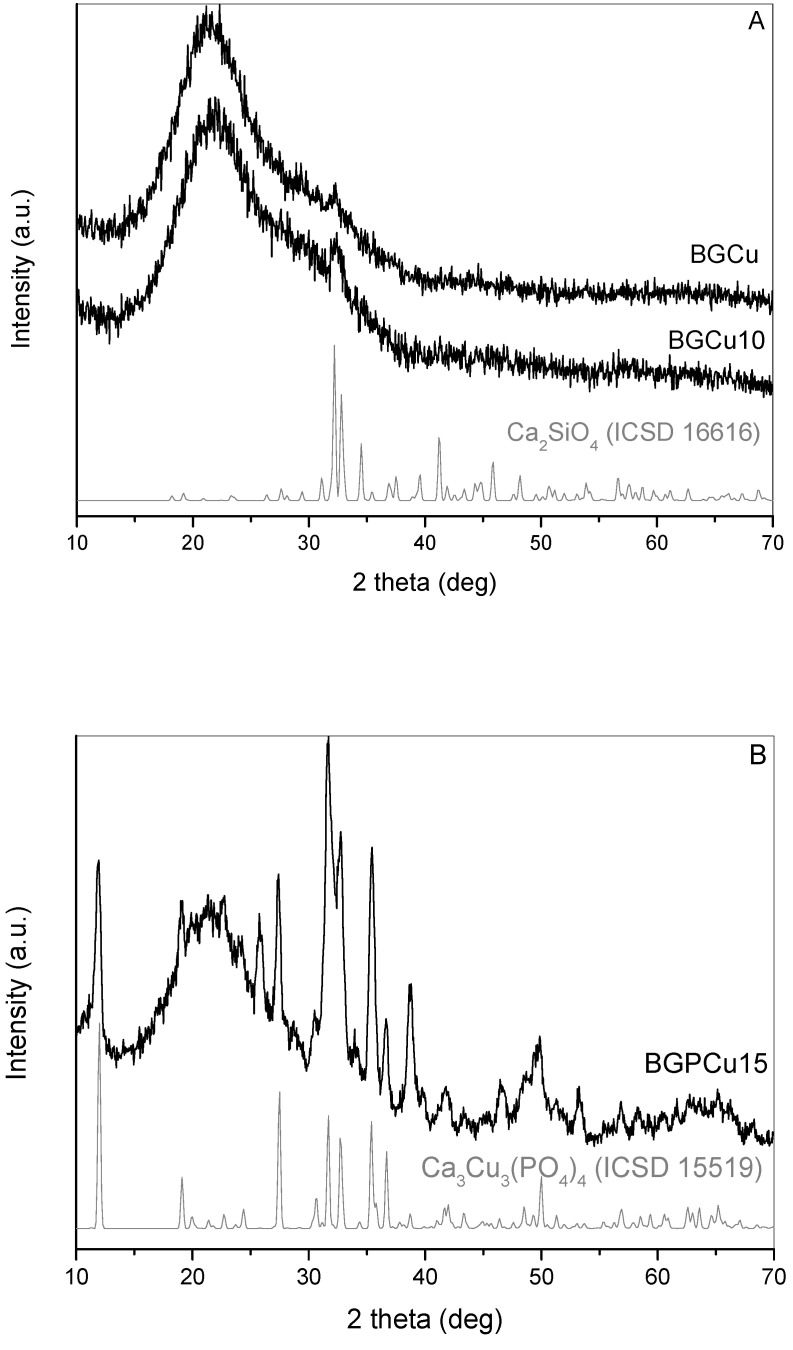
XRD of particles doped with Cu: (**A**) particles without P and (**B**) particles with P.

**Figure 4 nanomaterials-13-00165-f004:**
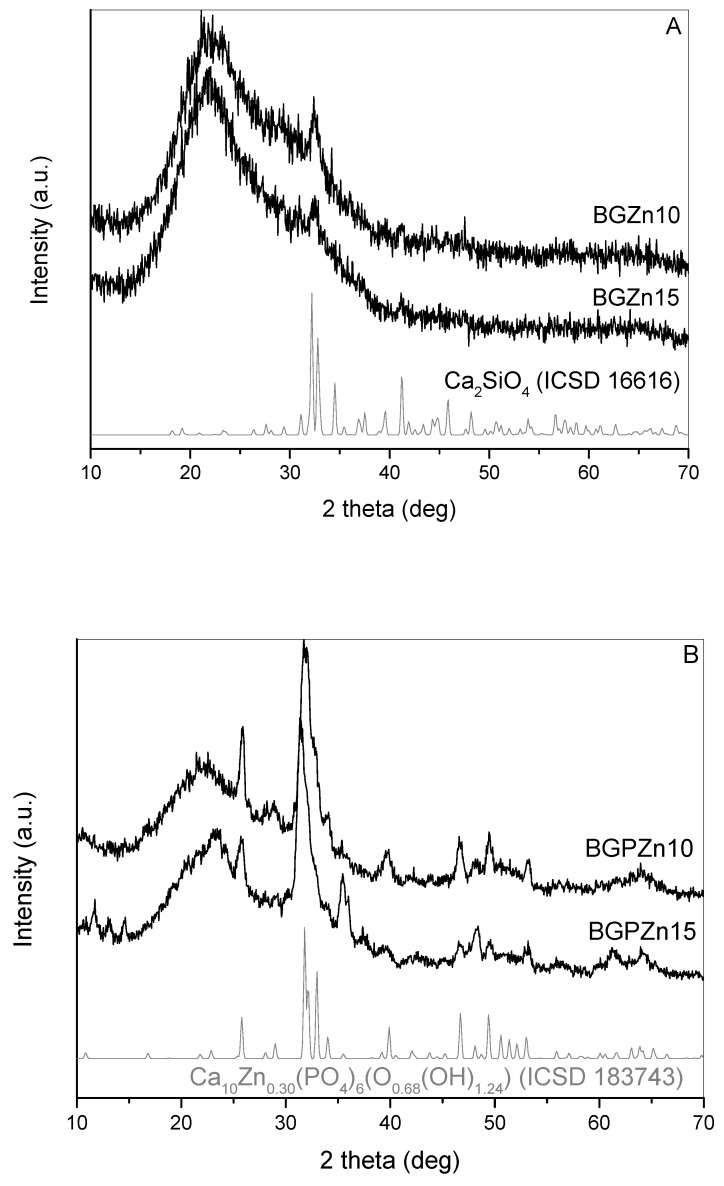
XRD of particles doped with Zn: (**A**) particles without P and (**B**) particles with P.

**Figure 5 nanomaterials-13-00165-f005:**
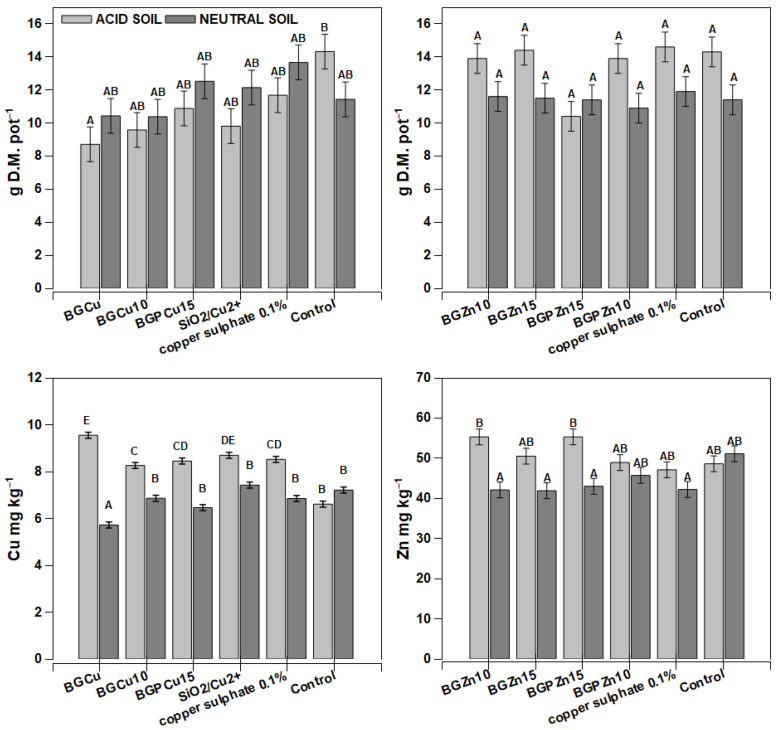
Yields of wheat *(Triticum sativum)* and concentration of copper and zinc after foliar application of different fertilizer formulations. Grey bars for acid, dark grey for neutral soil. Bars marked with the same letters represent homogenous groups after Tukey’s test at *p* < 0.05 and error bars represent standard error (n = 4).

**Figure 6 nanomaterials-13-00165-f006:**
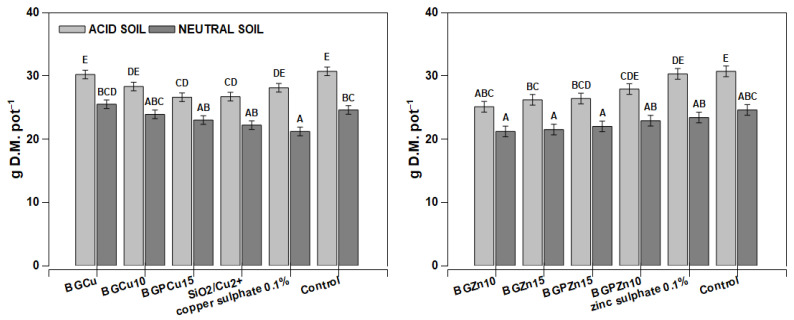
Yields of maize *(Zea mays)* after foliar application of different fertilizer formulations. Grey bars for acid, dark grey for neutral soil. Bars marked with the same letters represent homogenous groups after Tukey’s test at *p* < 0.05 and error bars represent standard error (n = 4).

**Figure 7 nanomaterials-13-00165-f007:**
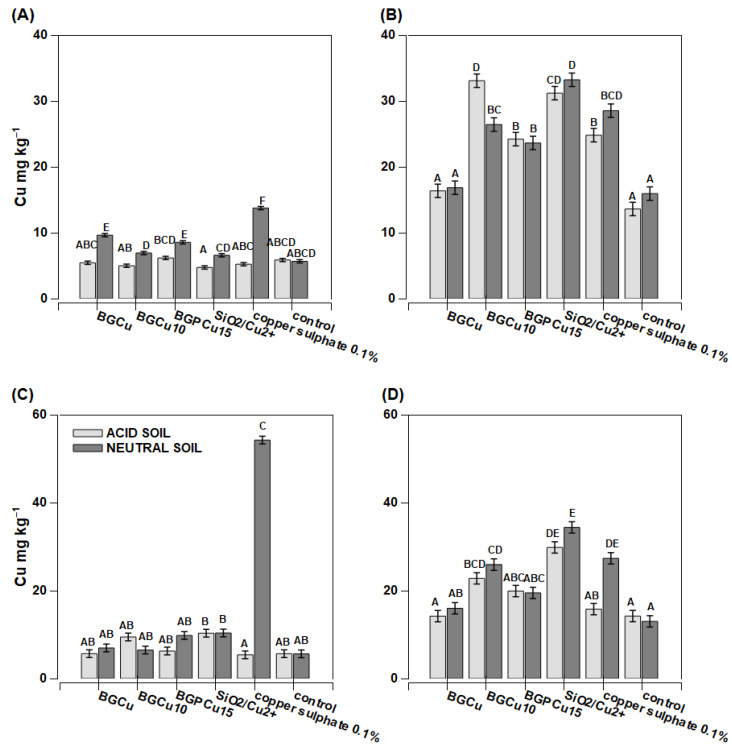
Copper concentration in maize *(Zea mays)* plant parts after foliar application of fertilizer. Grey bars for acid, dark grey for neutral soil Bars marked with the same letters represent homogenous groups after Tukey’s test at *p* < 0.05 and error bars represent standard error (n = 4). (**A**) In whole plant—preparation applied on one leaf; (**B**) in one leaf sprayed; (**C**) in whole plant sprayed; (**D**) in leaf cut from whole sprayed plant.

**Figure 8 nanomaterials-13-00165-f008:**
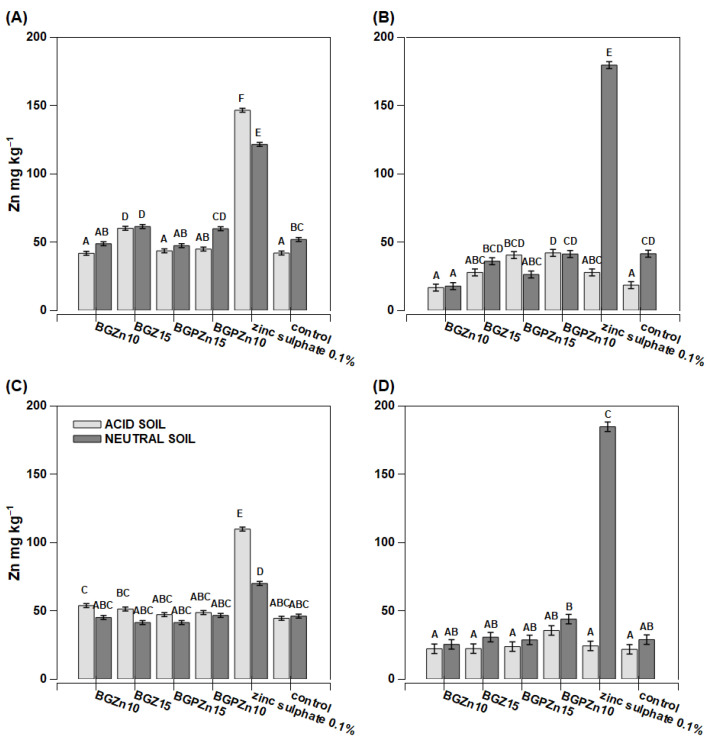
Zinc concentration in maize *(Zea mays)* plant parts after foliar application of fertilizer. Grey bars for acid, dark grey for neutral soil. Bars marked with the same letters represent homogenous groups after Tukey’s test at *p* < 0.05 and error bars represent standard error (n = 4). (**A**) In whole plant—preparation applied on one leaf; (**B**) in one leaf sprayed; (**C**) in whole plant sprayed; (**D**) in leaf cut from whole sprayed plant.

**Figure 9 nanomaterials-13-00165-f009:**
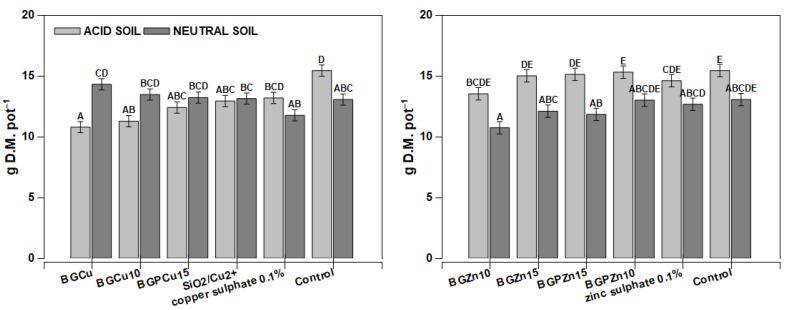
Yields of winter rape *Brassica napus* L. var *napus* after foliar application of different fertilizer formulations. Grey bars for acid, dark grey for neutral soil. Bars marked with the same letters represent homogenous groups after Tukey’s test at *p* < 0.05 and error bars represent standard error (n = 4).

**Figure 10 nanomaterials-13-00165-f010:**
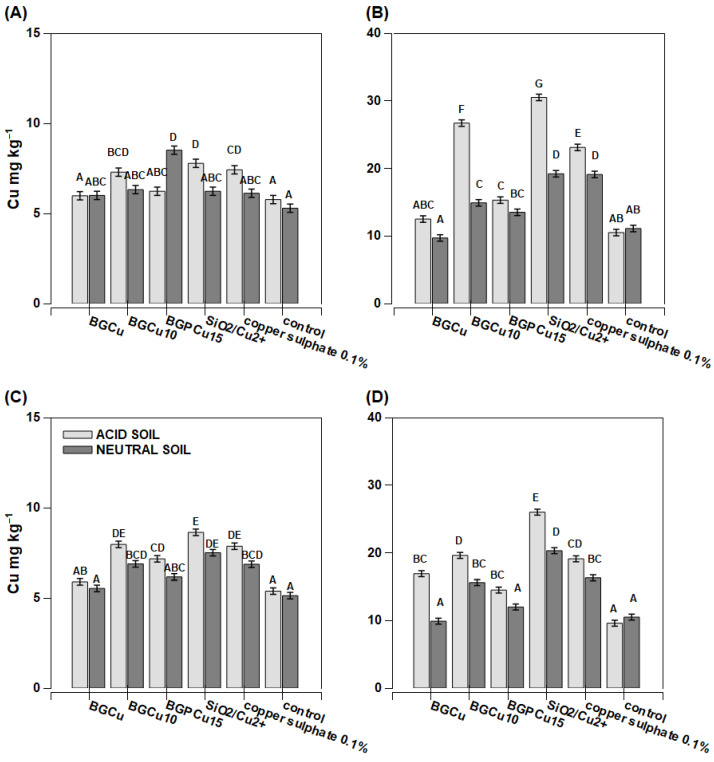
Copper concentration in winter rape *Brassica napus* L. var *napus* plant parts after foliar application of fertilizer. Grey bars for acid, dark grey for neutral soil. Bars marked with the same letters represent homogenous groups after Tukey’s test at *p* < 0.05 and error bars represent standard error (n = 4). (**A**) In whole plant—preparation applied on one leaf; (**B**) in one leaf sprayed; (**C**) in whole plant sprayed; (**D**) in leaf cut from whole sprayed plant.

**Figure 11 nanomaterials-13-00165-f011:**
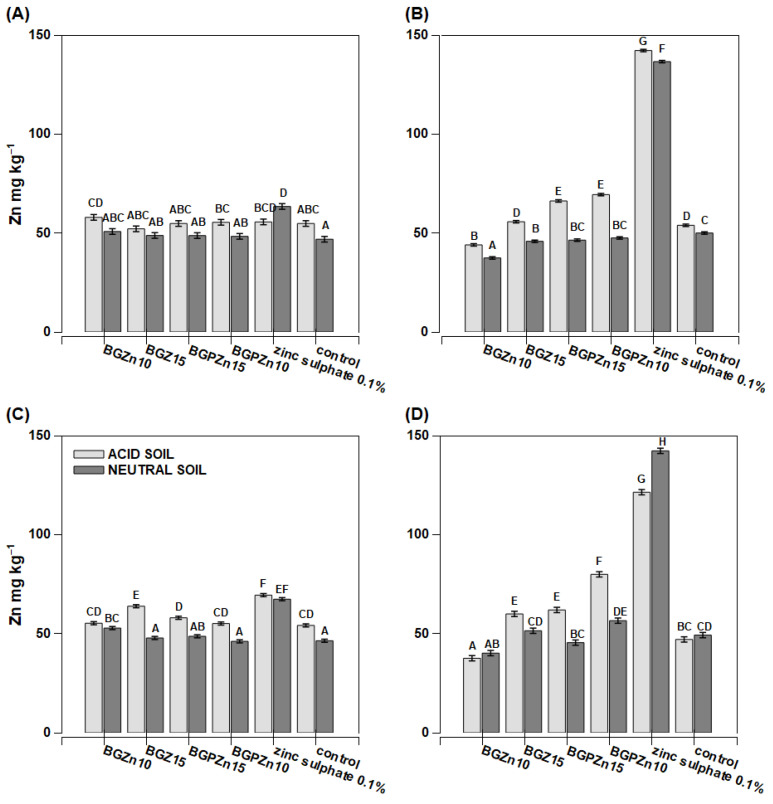
Zinc concentration in winter rape *Brassica napus* L. var *napus* plant parts after foliar application of fertilizer. Grey bars for acid, dark grey for neutral soil. Bars marked with the same letters represent homogenous groups after Tukey’s test at *p* < 0.05 and error bars represent standard error (n = 4). (**A**) In whole plant—preparation applied on one leaf; (**B**) in one leaf sprayed; (**C**) in whole plant sprayed; (**D**) in leaf cut from whole sprayed plant.

**Table 1 nanomaterials-13-00165-t001:** Ca, Cu and Zn content in the studied particles.

Sample	Ca g/kg	Cu g/kg	Zn g/kg
BGCu	55.40	41.62	-
BGCu10	131.21	103.26	-
BGPCu15	97.86	153.05	-
SiO_2_/Cu^2+^	-	150.64	-
BGZn10	131.29	-	105.81
BGZn15	99.76	-	155.06
BGPZn10	129.01	-	107.68
BGPZn15	97.64	-	158.99

**Table 2 nanomaterials-13-00165-t002:** Size and PdI parameters obtained by the DLS measurements.

Particles with Cu	Size (nm)	PdI	Particles with Zn	Size (nm)	PdI
BGCu	203	0.137	BGZn10	180	0.109
BGCu10	178	0.081	BGZn15	153	0.095
BGPCu15	179	0.112	BGPZn15	175	0.116
SiO_2_/Cu2^+^	202	0.065	BGPZn10	149	0.112

**Table 3 nanomaterials-13-00165-t003:** EDX analysis of the samples with Cu.

			Atom %		
Sample	O	Si	Ca	P	Cu
BGCu	74.40	20.54	2.14	-	2.92
BGCu10	73.72	22.40	0.93	-	2.95
BGPCu15	74.62	16.09	3.60	2.44	3.25
SiO_2_/Cu^2+^	75.59	19.92	-	-	4.49

**Table 4 nanomaterials-13-00165-t004:** EDX analysis of the samples with Zn.

			Atom %		
Sample	O	Si	Ca	P	Cu
BGZn10	72.32	22.60	2.87	-	2.21
BGZn15	72.71	22.77	1.52	-	3.00
BGPZn15	64.02	23.22	5.18	2.79	4.79
BGPZn10	67.86	15.99	8.83	4.16	3.16

## Data Availability

Not applicable.

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
