# Peer review of "Foliar Fertilization by the Sol-Gel Particles Containing Cu and Zn"

_nanomaterials, 2022, doi:10.3390/nano13010165_

Round 1

Reviewer 1 Report

In the present article, using sol-gel method, the authors synthesized silica particles (150-200 nm) containing Ca, P, Cu or Zn ions and tested as a foliar fertilizer on three plant species: maize Zea mays, 10 wheat Triticum sativum and rape Brassica napus L. var napus. The topic and results are interesting, and the conclusion summarizes the main points. However, some revisions are required before recommending the article for publication.

(1) For the SEM investigations, is it possible to include the operating voltage and currents?

(2) There are some typos in the report. Pls check the entire manuscript for typos and correct the wrong grammar.

Author Response

Dear Reviewer,

Thank you very much for the comment.

(1) For the SEM investigations, is it possible to include the operating voltage and currents?

This information has been included in section 2.3. Characterization of the particles in the form of the sentence: “The parameters selected in order to obtain a good quality images of the samples were: accelerated voltage 10 kV, emission current 67-74 µA and working distance in the range of 9-11 mm.” All samples were measured in the same conditions.

(2) There are some typos in the report. Pls check the entire manuscript for typos and correct the wrong grammar.

All text has been corrected.

Reviewer 2 Report

This article describes a method to increase the adsorption of Cu and Zn as foliar fertilizer respect to the conventional products as ZnSO4 and CuSO4 by the synthesis of nanomaterials supported on silica. The study highlights an increasing of uptake for the plant species analyzed, anyway several study are present in literature  (es. Le, V. N., Rui, Y., Gui, X., Li, X., Liu, S., & Han, Y. (2014). Uptake, transport, distribution and bio-effects of SiO2 nanoparticles in Bt-transgenic cotton. Journal of nanobiotechnology, 12(1), 1-15.)  that provide the direct evidence of the bioaccumulation of SiO2 nanoparticles in plants, which shows the potential risks of SiO2 nanoparticles impact on food crops and human health. In the article proposed this aspect has not been investigated. It is necessary to include this information in order to validate the proposed technology before we can submit the article. Moreover, additional information to SiO2 toxicity must be introduced both in the introduction and the discussion of the data acquired.

Author Response

Dear Reviser,

In some articles, SiO2 is indeed presented as a factor harmful to plants, but there are also articles showing a positive effect on plants, especially those exposed to environmental stresses (Ashkavand et al. 2015, Zarafshar et al. 2015, Yassen et al. 2017, etc ). Generally, Si is recognised as an element beneficial to plants which promotes their growth and is not harmful even in high concentrations (Mandlic et al. 2020, Pavlovic 2021). However its application as SiO2 requires further research.

Please be advised that the positive and negative aspects of using nano SiO2 have been mentioned in the Introduction and developed in the Discussion.

We also want to point out that our particles were not exactly nano, they were 150-200 nm in diameter. They were applied to the foliage and served more as a carrier for metal ions.

Our goal was to investigate in general whether synthesized NPs have any positive effect. Further detailed research should test the toxicity of all ingredients.

Ashkavand P, Tabari M, Zarafshar M, Tomášková I, Struve D. 2015. Effect of SiO2 nanoparticles on drought resistance in hawthorn seedlings. Forest Research Papers 76 (4): 350–359. DOI: 10.1515/frp-2015-0034

Zarafshar M., Akbarinia M., Askari H., Hosseini S.M., Rahaie M., Struve D. 2015. Toxicity assessment of SiO2 nanoparticles to pear seedlings. International Journal of Nanoscience and Nanotechnology 11(1): 13–22

Yassen A., Abdallah E., Gaballah M., Zaghloul S., Role of Silicon Dioxide Nano Fertilizer in Mitigating Salt Stress on Growth, Yield and Chemical Composition of Cucumber (Cucumis sativus L.). Int. J. Agric. Res. 2017, 12, 130-135. DOI: 10.3923/ijar.2017.130.135

Behboudi F., Sarvestani Z.T., Kassaee M.Z., Sanavi S.A.M.M., Sorooshzadeh A. Improving Growth and Yield of Wheat under Drought Stress via Application of SiO2 Nanoparticles. JAST 2018, 20, 1479-1492
http://jast.modares.ac.ir/article-23-19814-en.html

Kang H., Elmer W., Shen Y., Zuverza-Mena N., Ma C., Botella P., White J.C., Haynes C.L. Silica Nanoparticle Dissolution Rate Controls the Suppression of Fusarium Wilt of Watermelon (Citrullus lanatus). Environ. Sci. Technol. 2021, 55, 20, 13513–13522. https://doi.org/10.1021/acs.est.0c07126

Hu P., An J., Faulkner M.M., Wu H., Li Z., Tian X., Giraldo J.P. Nanoparticle Charge and Size Control Foliar Delivery Efficiency to Plant Cells and Organelles. ACS Nano 2020, 14, 7, 7970–7986, https://doi.org/10.1021/acsnano.9b09178
Yuvakkumar R, Elango V, Rajendran V., Kannan N.S. , Prabu P. Influence of Nanosilica Powder on the Growth of Maize Crop (Zea Mays L.). International Journal of Green Nanotechnology, 2011, 3, 180-190, DOI: 10.1080/19430892.2011.628581

Mandlik, R., Thakral, V., Raturi, G., Shinde, S., Nikolic, M., Tripathi, D. K., et al. (2020). Significance of silicon uptake, transport, and deposition in plants. J. Exp. Bot. 71, 6703–6718. doi: 10.1093/jxb/eraa301

Pavlovic J, Kostic L, Bosnic P, Kirkby EA and Nikolic M (2021) Interactions of Silicon With Essential and Beneficial Elements in Plants. Front. Plant Sci. 12:697592. doi: 10.3389/fpls.2021.697592

Reviewer 3 Report

Comments of nanomaterials-2120573

The main weaknesses of the manuscript:

1.     There is no need to show EDS spectra. Please replace by a table containing the quantitative analysis of all elements detected.

2.     Figs. 2-4: It is different from the normal XRD pattern and too messy, it is needed to improve.

Author Response

Dear Reviewer,

Thank you very much for the comment.

1. There is no need to show EDS spectra. Please replace by a table containing the quantitative analysis of all elements detected.

These tables have been inserted into the manuscript.

2. Figs. 2-4: It is different from the normal XRD pattern and too messy, it is needed to improve.

All XRD patterns shown in Figures 3 and 4 have been improved.

Round 2

Reviewer 2 Report

In accordance with the previous revision the article is ready for publication